# Communication and shared decision-making with patients with limited health literacy; helpful strategies, barriers and suggestions for improvement reported by hospital-based palliative care providers

**Ruud Roodbeen**[1,2]*, **Astrid Vreke**[1], **Gudule Boland**[3], **Jany Rademakers**[1,4], **Maria van den Muijsenbergh**[3,5], **Janneke Noordman**[1], **Sandra van Dulmen**[1,5,6]

**1** Nivel (Netherlands institute for health services research), Utrecht, the Netherlands, **2** Department of Tranzo Scientific Center for Care and Wellbeing, Tilburg University, Tilburg, the Netherlands, **3** Pharos, Dutch Centre of Expertise on Health Disparities, Utrecht, The Netherlands, **4** Department of Family Medicine, CAPHRI (Care and Public Health Research Institute), Maastricht University, Maastricht, the Netherlands, **5** Radboud university medical center, Radboud Institute for Health Sciences, Department of Primary and Community Care, Nijmegen, The Netherlands, **6** Faculty of Health and Social Sciences, University of South-Eastern Norway, Drammen, Norway

* r.roodbeen@nivel.nl

## Abstract

### Background

Communication and shared decision-making (SDM) are essential to patient-centered care. Hospital-based palliative care with patients with limited health literacy (LHL) poses particular demands on communication. In this context, patients' emotions and vulnerable condition impact their skills to obtain, understand, process and apply information about health and healthcare even more. If healthcare providers (HCPs) meet these demands, it could enhance communication. In this study, HCPs were interviewed and asked for their strategies, barriers and suggestions for improvement regarding communication and SDM with LHL patients in hospital-based palliative care.

### Methods

A qualitative interview study was conducted in 2018 in four Dutch hospitals with 17 HCPs—11 physicians and 6 nurses. Transcripts were analyzed using thematic analysis.

### Results

In general HCPs recognized limited literacy as a concept, however, they did not recognize limited *health* literacy. Regarding SDM some HCPs were strong advocates, others did not believe in SDM as a concept and perceived it as unfeasible. Furthermore, five themes, acting as either strategies, barriers or suggestions for improvement emerged from the interviews: 1) time management; 2) HCPs' communication skills; 3) information tailoring; 4) characteristics of patients and significant others; 5) the content of the medical information.

**Data Availability Statement:** All relevant data are within the manuscript and its Supporting Information files.

**Funding:** This work was supported by awards from ZonMw (the Netherlands Organisation for Health and Research Development) project number 844001403 to authors RR, GB, JR, MvdM, JN, and SvD. The funders had no role in study design, data collection and analysis, decision to publish, or preparation of the manuscript. No commercial companies funded the study or authors.

**Competing interests:** The authors have declared that no competing interests exist.

## Conclusions

According to HCPs, more time to communicate with their patients could resolve the most prominent barriers emerged from this study. Further research should investigate the organizational possibilities for this and the actual effectiveness of additional time on effective communication and SDM. Additionally, more awareness for the concept of LHL is needed as a precondition for recognizing LHL. Furthermore, future research should be directed towards opportunities for tailoring communication, and the extent to which limited knowledge and complex information affect communication and SDM. This study provides first insights into perspectives of HCPs, indicating directions for research on communication, SDM and LHL in hospital-based palliative care.

## Introduction

Effective communication, shared decision-making (SDM) and assessing the needs of patients and their significant others during palliative care is an important goal in medical practice. According to the World Health Organization, palliative care is defined as: "an approach that improves the quality of life of patients and their families who are facing problems associated with life-threatening illness" [1]. Each year an estimated 20 million people need palliative care worldwide [2]. In the Netherlands, approximately 120,000 people per year receive palliative care before they die [3]. During this period, many people receive hospital care, which is often highly complex and preference-sensitive [4]. Therefore, SDM in palliative care is important.

Many patients in the palliative phase experience problems communicating with their health care providers (HCPs), such as not understanding or using the provided information [5], or a limited recall of medical information [6,7]. These problems might be attributed to the emotional and psychological distress caused by the patient's current medical situation [8]. Other contributing factors could be a lower level of education, a higher age of the patient or perhaps, limited health literacy (LHL) of the patient [5,9–11]. Having difficulties to communicate with HCPs is problematic, because patients consider good communication with their HCP as one of the most important factors in palliative care [12]. Furthermore, poor communication can lead to patient dissatisfaction, discontinuity of care, diminished patient safety and autonomy and hampers SDM [13].

SDM is a renowned method for making decisions with patients in healthcare settings. It enhances patient autonomy and patient-centeredness.[14,15]. Within the context of palliative medicine, SDM implies that HCPs inform their patients about the different treatment options available, the possibility of refraining from treatment, and the outcome of these different options [16]. Furthermore, the patient and the HCP discuss personal preferences of these different options and eventually take a decision together [16]. In order to achieve SDM, effective communication between HCPs and their patients is required. In palliative care, most patients also want to participate in making decisions about their medical and psychological care to some extent [17], while a minority prefers decisions to be made by the HCP [18]. Yet, despite the importance and need for SDM, this is not always accomplished [19,20].

Considering the challenges and difficulties of making complicated decisions on itself in palliative care, it is an even bigger challenge for patients with LHL [21]. Health literacy is defined as having the skills to obtain and understand information about health and healthcare and the ability to put this information into practice [21,22]. LHL is predominantly observed in illiterate

people, people with little or low education, the elderly and in the first generation of non-western migrants [22–24], but can be observed in highly educated people as well. LHL is estimated to be present in 36% of adults in the Netherlands [11]. LHL could lead to adverse health outcomes, LHL patients spend more time in hospital and have lower adherence to medication and treatment [9]. It also complicates communication which reduces the possibility for SDM [10]. Patients with LHL participate less in SDM with their HCPs compared to health literate patients [11].

Health literacy is increasingly gaining attention, because it is one of the central determinants of inequality in healthcare [25]. Nevertheless, few studies have investigated LHL in palliative care focusing on communication and SDM using qualitative methods [26]. What we do know to be effective are communication strategies aimed at improved communication between HCPs and patients with LHL. For instance, the teach-back method', chunk and check and using pictures and illustrations [26–30]. In addition to these strategies, exploring perceived barriers and suggestions of HCPs are important, because they could help assess the implementation and/or feasibility of strategies. Little is known about strategies, barriers or suggestions HCPs have with regard to communication and SDM in hospital-based palliative care, and especially so with patients with LHL. For this reason, the aim of this study is to investigate exactly that. Insights could offer HCPs recommendations to adjust their communication to the needs and preferences of patients, checking understanding and enhancing recall of medical information by patients.

## Materials and methods

### Research team and reflexivity

All interviews were conducted by one male researcher (RR), occupied as a communication researcher (M.Sc.) at the time of the study, and experienced in conducting qualitative in-depth semi-structured interviews. The researcher was trained by coauthors (MvdM, JN, SvD) to conduct interviews with HCP in palliative care (e.g., how to prevent HCPs from straying from subjects or questions and communicate effectively).

The researcher had no profound prior relationship with participants. Furthermore, participants were aware that the researcher was not medically trained or involved in patient care, and participants generally knew the goals and reasons behind the interviews (i.e., investigating their roles as HCPs in palliative care regarding communication, their patients and the organization of their hospital).

### Study design

The theoretical framework underpinning this qualitative interview study is based on a phenomenological approach, focusing on describing meaning and significance of experiences. This design enables participants to express the barriers they experience regarding communication and SDM in their own words, and to say how they think it could be improved [31]. Apart from being helpful for other HCPs, the interviews also intended to generate knowledge for developing an online communication training as part of a larger study on palliative care with patients with LHL called 'A basic understanding'—see S1 Appendix.

All participants were selected using convenience sampling. The participating hospitals appointed an employee as a project manager functioning as a 'point of contact' for the researchers—in most cases a specialized nurse working in the department—and this person invited other HCPs face-to-face or via email. The HCPs included were physicians and nurses who regularly conduct consultations with patients with cancer or chronic obstructive pulmonary disease (COPD), and discuss palliative care and treatment options. 21 HCPs were invited

to participate, four did not respond to the invitation (no reasons provided). Interviews with 17 HCPs allows for a diversity of perspectives to profoundly assess strategies, barriers or suggestions of LHL in palliative care focusing on communication and SDM.

As part of 'A basic understanding', HCPs from four Dutch hospitals participated in the interviews. The hospitals were located in different regions in the Netherlands; three were university hospitals and one a general hospital. HCPs worked for the palliative care, pulmonology, radiotherapy, oncology or anesthesiology departments. Interviews with HCPs took place at the hospital where they work. During all interviews, only the researcher and participant were present.

The interviews were semi-structured and conducted with a topic list. All interviews were carried out between April and October in 2018. An initial version of the topic list was developed based on literature and experience from previous research with cancer patients [32] (JN). Feedback on this initial version was provided by researchers with ample experience in research focused on LHL (GB, JR & MvdM). After pilot-testing this version in the field, and after some minor adjustments, the topic list was completed (RR)—it can be found in S2 Appendix. No repeat interviews were conducted, no extensive field notes were recorded during or after the interviews. Informed Consent was signed for by the HCPs. All interviews were audio-recorded and transcribed verbatim by an external transcription service. The interviews took on average 46 minutes, ranging from 33 to 70 minutes. Three were excluded in this calculation, since the duration was accidentally not recorded by the researcher (RR). To increase the credibility of the results, a member check of transcripts was performed—in which the transcripts were given to the participants in order to check the authenticity of the transcripts. Participants did not provide feedback on the results of our study.

## Analysis

Transcripts were analyzed using thematic analysis following the phases described by Braun and Clarke [33]. For the purpose of the present study, the analyses focused primarily on the data gathered with the questions in the third section of the topic list (see S2 Appendix). Two coders coded the data and, to identify initial and preliminarily themes in the material, read the first 10 transcripts, generating, discussing and reviewing initial and preliminarily codes (RR & AV). These initial themes were all derived from the data and reviewed and named, following an iterative pathway [33]. Subsequently, transcripts were imported in MAXQDA11 and coded by one researcher (AV). To increase reliability, investigator triangulation was applied: ten of the interviews were additionally coded by another researcher (RR). The themes and subthemes that emerged during the analysis were discussed among three researchers (SvD, RR & AV), who then came to an agreement on themes. By analyzing segments and codes within themes, one researcher (RR) finalized the naming, positioning and describing of (sub)themes and completed the analyses.

A coding scheme was created (Table 2), in which themes, sub-themes and elements within sub-themes were presented. All (sub)themes that emerged during the thematic analysis are illustrated by multiple quotes in the results section of this study, which were translated into English and edited, increasing readability without the loss of meaning or context.

## Ethical considerations

To protect the privacy of the participants, their records were anonymized and all data that could reveal the identity of the participants were deleted from the transcripts. After completing all member checks of the transcripts, the audio recordings were deleted. The study protocol

was evaluated by the Medical Ethical committee of the Radboud university medical center, which exempted the study from formal ethical approval (filenumber CMO: 2017–3623).

# Results

## General results

A total of 17 HCPs from 4 different hospitals—3–5 per hospital—were interviewed. Table 1 shows an overview of HCPs' characteristics, 6 nurses and 11 physicians participated, 10 were female. One physician was in training at the time of the interview.

At the start of the interview, HCPs recognized limited literacy as a concept, however, they had never heard of limited *health* literacy. During the interview they provided different interpretations for the concept compared to the concept described in the introduction of this study. After the interviewer (RR) provided them with this definition, all HCPs reported to have had experience with patients with LHL in their consultations. Regarding SDM, the opinions of HCPs varied widely. While some HCPs were strong advocates, others did not believe in SDM as a concept, caused by the, in their view, irreconcilable HCP-patient imbalance in medical knowledge and experience. Some HCPs believed that patients with LHL would be unable to comprehend the different treatment options and would have to be directed towards a certain decision. However, a few HCPs considered being directive during SDM only acceptable when patients make decisions that are unrealistic or pointless, or when patients are incapacitated and/or have no legal representation—i.e., legal guardians who are appointed to make decisions when patients are not capable to do so (see Quote 1).

*Quote 1. Oncologist.*

*Respondent: . . . I think it is very important, as a doctor, to leave out your own opinion a little bit. Of course, as a doctor, you know very well what is and what isn't useful or meaningless. Therefore, I will not consciously talk my patients into a meaningless treatment, but I can discuss the pros and cons of a treatment with the patient. If the patient says, "for me, three months of extra time is not at all important, I just don't want to go to the hospital", maybe as a doctor, I think and feel that three months is very important. As a doctor, I can have a different opinion about that, but if this patient clearly indicates, "I know I am not getting better and I think the extra three months is not enough to start this treatment", then that's fine for me, if the patient knows what the pros and cons are. But I'm not going to support a patient not to do a simple life-saving treatment.*

**Table 1. HCPs' characteristics (n = 17) at the time of the interview.**

| Profession | n | % |
|---|---|---|
| Nurses | 6 | (35) |
| Physicians | 11 | (65) |
| **Departments** | | |
| Palliative care | 5 | (29) |
| Pulmonology | 5 | (29) |
| Radiotherapy | 3 | (18) |
| Oncology | 3 | (18) |
| Anesthesiology | 1 | (60 |
| **Sex** | | |
| Female | 10 | (59) |
| Male | 7 | (41) |

Some HCPs believed that SDM is especially applicable in palliative care, since there is more clinical equipoise—i.e., balance of forces and interests—in palliative than in curative care.

## Themes emerging from the analyses

An overview of (sub)themes and elements are presented in Table 2.

**1. Time management.** One of the most prominent themes emerging from the interviews as a strategy and barrier, was time management. HCPs considered more time for their consultations as a necessity to resolve all existing communication barriers. In general, consultations of around 30 minutes were considered appropriate by HCPs, instead of the 10 minutes that are currently standard practice in hospitals. Despite HCPs' ability to, for instance, book double-time consultations as a strategy, or manually expand visiting times in their schedule, some also mentioned experiencing a tension between adding time and the organizational and financial constraints of hospital management (Quote 2).

*Quote 2. Physician pain and palliative medicine.*

*Respondent: . . . Look, as an example, on my outpatient clinic consultations, every twenty minutes, I see another patient. If I see a patient with high health literacy, who, prior to our consultation, has read everything on Google, and we agree on treatment and are able to engage in good SDM, then a consultation will take up ten minutes. However, if I have a patient who does not prepare for the consultation and does not understand the information, and brought three significant others with him who don't understand the information either, in that situation, twenty minutes is too short. The system won't allow me to say, "well guys,*

**Table 2. (Sub)themes emerging from the analyses regarding HCPs' strategies, barriers and suggestions for improvements in communication and SDM with patient with LHL.**

| Themes | Sub-themes | Elements |
|---|---|---|
| **1. Time management** | Limited time | - practical or organizational barriers |
| | | - substantive barriers |
| | | - executing SDM |
| | Additional time | - provide information |
| | | - forming a relationship—bonding |
| | | - lengthening the SDM process |
| **2. HCPs' communication skills** | Observing and assessing LHL | |
| | Tools & aides | |
| | Limited skills | - adjusting to LHL |
| | | - unilateral outlook on treatment/care by HCPs |
| | More collaboration | - between HCPs and disciplines |
| | | - between colleagues in the hospital |
| | Education | - communication skills |
| | | - the sharing of experiences |
| | | - additional tools supporting HCPs during and patients prior to the consultation |
| **3. Tailoring** | Simplifying | |
| | Sensitizing | - comprehensive and supportive interaction |
| | Repeating | |
| **4. Characteristics of patients and significant others** | Knowledge | |
| | Attitude, mood or condition | |
| | Language and culture | |
| **5. Content of medical information** | | |

*let's pull the emergency brake, stop my train and start working with these people". I could offer these people an additional appointment in two weeks, in which we have more time for more explanation. However, if I do this too often, if I do this with more than ten percent of my patients—scheduling additional appointments, this will displease my director. He will tell me, "you do not reach the potential of your production". So, there is a field of tension in this situation. . .*

Limited time results in *practical or organizational barriers in communication*. Related to patients with LHL, and according to the HCPs, explaining medical information takes up more time, and therefore forces them to select the information they perceive as relevant. Because of limited time, a thorough execution of the teach-back method is not considered realistic with LHL-patients. Furthermore, limited time withholds HCPs to engage in in-depth interactions with their patients, causing a more *substantive barrier in communication*. According to HCPs, effective communication and SDM is only achievable if there is a level of understanding or 'bond' with the patient and their significant others. Also, HCPs stated that bonding takes time and, in most cases, multiple encounters are required. According to HCPs, limited time causes inabilities to assess or talk about health literacy and about the psychological, social and/or spiritual/existential aspects of care. It also withholds them from profoundly assessing and discussing palliative care related to the quality-of-life preferences most important for patients. Limited time also causes barriers in *executing SDM*. If time is limited, SDM around difficult and life-changing decisions is not always feasible. According to one HCP, an example of this is when the start of treatment cannot be delayed. For instance, in small cell lung carcinoma, delaying treatment could negatively alter survival rates.

*Additional time* provides HCPs with the opportunity to *provide information*. It would allow them to inform the patient about their disease and to effectively engage him or her in SDM. Additional time also offers HCPs the opportunity of *forming a relationship (bonding)* with the patient. HCPs suggest that time allows for important informal conversations, for instance, to engage in small talk and to get to know the person behind the patient. According to HCPs, this allows for longer and more elaborate conversations about quality of life, and in general, facilitates the possibility of finding a moment of serenity and see patients holistically (Quote 3).

*Quote 3. Specialized nurse in oncology.*

*Interviewer: . . . And what would help to make that ['that' means room for bonding] clear to patients?*

*Respondent: Clarity and honesty, I think, from the very beginning. And for the caregiver, I think that if you can, you must give yourself time and peace of mind to see the patient as a whole. Use something like a time-out, which in the Netherlands of course happens during a lot of different treatment processes. If you provide information to a patient, you schedule a time-out and offer patients time for reflection, in order for the patient to come up with questions, or have discussions with their significant others, so that you only make the decision in a second conversation with the patient, after the time-out.*

Furthermore, some HCPs suggest *lengthening the SDM-process*. According to some HCPs, the process should start at the beginning of the care trajectory, and not just when patients have to decide. Preferably, patients should be allowed plenty of time and space to make decisions, for instance, by scheduling a separate consultation for discussing options or confirming decisions.

**2. HCPs' communication skills.**   HCPs recognized limited literacy as a concept. And although they had never heard of limited *health* literacy, they did describe communication strategies that helped them to recognize LHL. HCPs closely *observe and assess* the appearance, non-verbal communication and language of patients and significant others. HCPs suspect LHL when patients enter the consulting room seemingly unmotivated and uninterested, look at handed-out leaflets or medical forms with a dazed look or dress sloppily, having an unkept appearance. Furthermore, an overly simplified description of their medication, the choice of words when asking and responding to questions, the capacity and coherence of questions asked by the patients and the quality of the responses by patients on teach-back questions are seen as indications of LHL. However, despite these indications, some HCPs do encounter patients who are capable to conceal their limited skills and pretend to understand information. As a more direct approach, some HCPs reported to ask for the occupation of the patient. HCPs are reluctant to ask for the level of education, since patients don't automatically understand the legitimacy for this question (Quote 4).

> *Quote 4. Oncologist.*
>
> *Interviewer: And during conversations with patients, how do you recognize LHL?*
>
> *Respondent: I always try to let patients explain in their own words what [disease] they have, what they already know and what they have heard so far. Then you have at least some insight into the words someone uses and what information someone has received. For instance, the questions I ask when someone is forwarded by the surgeon, "What is it that the surgeon already discussed with you?", "What was the main reason for coming here?", "Is it clear to you what the purpose of this conversation is?" This is a point at which some patients already lose their head. Then I start by explaining who I am, what an oncologist does and what they can expect from me during the conversation. I'm not going to start with, "gosh, what is your level of education?" That doesn't have to be related to health literacy at all.*

Lastly, HCPs base their assessment on the medical and lifestyle-adherence of patients and, after consoling with a colleague, on the opinion of other HCPs.

As a strategy using *tools and aides*, HCPs reported to use materials for patients to take home (e.g., handing out leaflets or writing checklists for patients) and materials that support their explanations (e.g., using or drawing pictures, videos and anatomical models) or use the internet to search for information regarding the living environment of the patient (Quote 5).

> *Quote 5. Specialized nurse palliative care.*
>
> *Interviewer: So, by 'playfully', you mean that you loosely sense it during the conversation?*
>
> *Respondent: Yes. That's right. Often, my start of the conversation, when I know I'm seeing people, I obviously look at what they have [their disease], but also, look at where people live. If I know their address from our system, then I visit Google Maps and look up their address. Using Street View, I observe their general environment, their street, their garden, is there a park nearby, is there a lake nearby? . . . I really use that [during conversation]. For instance, if people say, "gosh, I really like walking", then I say, "then you probably will visit the lake that lies there and there". Then they feel understood and they don't feel like a number or a patient anymore. Instead, they think, "hey, this guy knows what my surroundings look like" . . . I always try to 'level' with the patient. In my experience, it is an entrance to [. . .] reaching a very nice level, where people can talk to you easily.*

HCPs indicated that they only use visual tools after recognizing LHL. Furthermore, some of the HCPs reported to use the teach-back methodology as a tool to check if patients understand the information, however, they do experience that some patients feel 'tested' or 'dumb' when asked this question. Therefore, HCPs indicated to be reluctant to engage in teach-back. According to the HCPs, a solution for this is to summarize the information and ask the patient for a confirmation. Some HCPs also use the patients' significant others as aides to confirm the information the patient is telling and ask them to help the patient with understanding the information and making decisions. They argue that if a significant other understands the information and options, they can help the patient at home. However, not every HCP is pleased with the presence of significant others. A few HCPs reported that families can become dominant, making it difficult for them to elicit the actual preferences of the patient. Further-more, HCPs indicated to use other professionals as aides, to use interpreters during consulta-tions, or their own secretaries providing additional information to LHL-patients over the phone. Fellow HCPs are used for exchanging experiences in communication or for referring patients. In addition, HCPs use their own computer systems as a tool, discussing topics using 'smart phrases' in order to add structure to their consultations. After the consultation, and if HCPs identify LHL or potential misunderstandings, they schedule a call-back, an additional appointment, or reschedule their subsequent appointment to the end of the day.

Important barriers in communication are the *limited communication skills* of HCPs them-selves, and their inability to *adjust to LHL-patients*. For instance, HCPs find it hard to make complicated medication-information understandable to patients, information they need to participate in decision-making. Although HCPs actively try not to use medical jargon and complicated words, they still feel that they are unable to solve the imbalance that exists between their own knowledge and the knowledge of the patient. Furthermore, some HCPs reported they (or their colleagues) can have a *unilateral outlook on treatment*, meaning that HCPs have a strong tendency towards treating patients. As a consequence, treatment-options are framed—e.g., barely mentioning the option of withholding treatment or waiting. According to these HCPs, this could create unrealistic expectations and a sense of unnecessary urgency for patients having to undergo treatment. In addition to a unilateral outlook on treatment, some HCPs reported to have a *unilateral outlook on healthcare* in general, and only discuss the phys-ical aspect of care with the patient. According to these HCPs (Quote 6), they sometimes hide behind medical technicalities, inhalers and pills to avoid a 'real talk'.

*Quote 6. Pulmonologist.*

*Interviewer: To what extent are these different aspects covered during conversations with patients?*

*Respondent: Very few. Much too little. Yes.*

*Interviewer: Oh, very few. Meaning all aspects, except the physical?*

*Respondent: Yes, we do talk about the technical aspects, the puffs and the pills and the oxygen levels and the physical therapist, and maybe even the lung function numbers or blood results. But just the question, "what do you actually expect from me", or, "what do you think about how things will go in the coming year", or, "will you still be here in a year", or "do you expect to be there in a year"? . . . The eight-minute consultations we now have—my consultations last eight minutes on average—are not suitable for this. . . . The moment you ask such a ques-tion, you already know that it will take a while. However, you should actually do that. In fact, you should say [to the patient], "I think your condition is deteriorating, you notice this as well, we should think about the near future". Sometimes you do that [ask such a question],*

*and sometimes it works. Occasionally, on their own initiative, a patient comes up with questions like this, but there aren't many people who are wondering, "how am I actually doing?" One of the problems is that the patient does not come to the consultation prepared at all.*

In addition, some HCPs indicated that they don't talk about spiritual matters or existential questions, because they lack the know-how or the life-experience, or they don't believe that it is their role to talk about these subjects with patients. In contrast, some HCPs point out that talking about spiritual care or existential questions is an important part of being an HCP in palliative care (Quote 7).

*Quote 7. Specialized nurse palliative care.*

*Respondent: 'Philosophy of life' and 'cultural background' are now placed all the way down at the bottom [the interviewer had asked the respondent to sort out cards with palliative care topics]. However, with this [placing these cards last], I'm not saying that it isn't our job to talk about these subjects, because I think it actually is. It is our job. Although we sometimes, but not enough, raise these subjects in conversation, I also think it is our task, but this applies to all of the care providers involved. So, not only for a doctor, but also for a nurse, and a social worker, and a dietician.*

Furthermore, some HCPs indicated that discussing the more profound aspects of care (e.g., talking about spiritual care or existential questions) is easier with patients with a low level of education, because they dare to be more vulnerable and 'open-up' more easily (Quote 8).

*Quote 8. Internist and consultant palliative care.*

*Interviewer: . . . And is bringing up aspects of care by yourself [physical, psychological, social and existential/spiritual] the same when people have a low level of education? Is that the same?*

*Respondent: Yes, that's the same thing. Then it might be even easier.*

*Interviewer: Okay. Why?*

*Respondent: Because they [low educated people] often talk easier. That obviously doesn't apply to everyone. However, I do think that people with a higher education keep more to themselves. And with lower educated people it is easier to talk, "gosh, what makes it all worth to you?" "What do you do all day?" Yes, that is easier.*

*More collaboration between HCPs* in the hospital *and other disciplines* involved in health care was mentioned as a suggestion for better communication, especially around the organization of care. According to HCPs, all information must be accessible for patients and their significant others, between HCPs in the hospital and other disciplines (e.g., primary care). HCPs suggested that improving collaboration with general practitioners (GP) could improve communication, since, in the Netherlands, the GP is the first point of contact for many patients, and in general has a lot of contact with palliative patients. A few HCPs suggested to ask patients about their relationship with their GP before transferring information. If this relation is compromised, this could increase miscommunication and, according to these HCPs, possibly even risk unnecessary hospital admission. According to some HCPs, the ideal way of transferring patients—from the hospital to a home situation—is in a consultation with both the GP and patient present in the same room. However, these HCPs do admit that this is practically and financially unrealistic. Lastly, increasing collaboration *between colleagues in the hospital*

could improve communication. In addition to their strategies, some HCPs suggest that they, as a team or department, also should have a communication-strategy meeting for specific LHL-patients, aligning the way they talk to them, what they say and when they say it (Quote 9).

*Quote 9. Specialized nurse palliative care.*

*Interviewer: Do you also experience barriers in providing care for patients with LHL?*

*Respondent: Yes. Sure. I think that if these patients have multiple HCPs, it is difficult to align the way all of these HCPS communicate with these patients.*

*Interviewer: Do you mean, for example, that the doctor should communicate in the same way as you do?*

*Respondent: Yes. Yes. That we, before talking to the patient, should discuss the things we are going to explain to the patient, and how. And that you, only after discussing this, should approach the patient. However, this does not happen, of course not. The bustle and crowded-ness of the hospital makes that . . . First, the doctor visits the patient and tells this . . . Then I come in and tell the patient that . . . I think that if you first have a multidisciplinary gathering and agree about how to approach such a patient, that could result in a patient who better understands the information, and therefore could be more loyal to his therapy. Yes.*

Additionally, HCPs should not hesitate to transfer patients to other HCPs within the department if communicating is difficult. According to some of the HCPs, without having an explanation or reason for this, sometimes, communication just does not work between an HCP and patient (i.e., there is no 'click' or 'connection'). These HCPs suggested that a transfer to another HCP could offer the patient a fresh perspective and could potentially improve communication.

Furthermore, HCPs suggested *education* for increasing their own *communication skills*, *the sharing of experiences* between colleagues, and the availability of *additional tools supporting HCPs during and patients prior to the consultation*. Some HCPs suggested to have peer-meet-ings solely dedicated to exchanging real experiences between colleagues regarding communi-cation with LHL-patients. Also, in these meetings, video-recordings of real consultations might be helpful to reflect on their own skills. According to these HCPs, to increase awareness for LHL in these meetings, emphasis should be on the limited amount of information LHL-patients are able to process. Although most HCPs reported to use all kinds of tools or aides as a strategy to support their explanations (e.g., drawing pictures) or provide information for patients to take home (e.g., handing out leaflets or checklists), they indicated needing addi-tional tools. To support their explanations during the consultation, some HCPs mentioned requiring templates of the human body and/or organs for them to write on as a tool for explaining the disease, and suggested to develop educational videos that explain to patients how treatment is administered to them. To support patients, some of the HCPs suggested to use tools to prepare patients for their consultations and ask them questions using surveys prior to the consultation. However, these HCPs do understand the difficulty of filling in these ques-tions for LHL-patients. Furthermore, HCPs would like to receive tools for recognizing LHL, and receive information and training on how to use these skills during communication and SDM (Quote 10).

*Quote 10. Specialized nurse in oncology.*

*Interviewer: What would you need to be able to align healthcare to patients with LHL?*

*Respondent: Well, a nice thing is that we sometimes have additional training, also in that area. That makes everything livelier again. You can exchange tips, asking, "how do you deal with that?" For example, drawing for people, drawing a schedule indicating this week you must do this, and this week you must do that. . . . Or drawing wounds for people, showing them how it will look. Or for people with gynecological cancer, drawing what it looks like from underneath, because that will look very strange.*

*Interviewer: Because then you learn to use tools that help to explain better?*

*Respondent: Yes. [. . .] I don't mean the drawing in itself obviously, but for example a tip, if you notice that people do not understand your explanation, visualizing is useful. I have learned that from lessons about people with incapacities in "health issues". Continue to share [experiences and tips] with each other in lessons. [The trainer should be] someone who has devoted himself to this, who is an expert in it, a psychologist or a nurse or a doctor. Every nurse, every doctor, has to deal with this. It's nice to share things with each other. Then it remains fresh and you think, "oh yes, those people [patients with LHL] sometimes need different methods, let's try them out. Yes. That is always nice to do, additional training.*

**3. Tailoring.** Based on their assessment of LHL, HCPs report to tailor their communication. Yet, despite using strategies, some patients are just not ready to deal or talk about palliative care or dying (Quote 11).

*Quote 11. Specialized nurse palliative care.*

*Interviewer: . . . Are there patients with whom you find communication difficult?*

*Respondent: Yes. Some people just don't like talking about death or about illness. [. . .] For example, [. . .] someone who says, "well, talking about death, that is no fun, so instead we are going to have a coffee". That's difficult. Seemingly, some people don't want to talk about it and need their own approach. Well, you have to accept that people don't want to talk about it, at least at that moment, and that they need their own strategy. Sometimes you haven't found that strategy yet.*

HCPs reported to often intentionally *simplify* their language, use short sentences and easy language, avoid jargon, talk slowly, et cetera. They also limit the information they share with their patients and alter the ways in which they ask questions. They withhold overly detailed or complicated information, use only examples that are relevant for the patient or only ask the patient closed-ended questions. HCPs do recognize that simplification bears risks, because it may lead to incomplete and less specific information, and could implicitly be directive towards a decision (Quote 12).

*Quote 12. Pulmonologist.*

*Interviewer: How do you try to take this group of patients [patients with LHL] into account when communicating in general?*

*Respondent: Yes, by outlining it more simple, or by using more simple language . . . However, I often think that despite this, we still talk too complicated. Even for patients without LHL . . . Sometimes it doesn't even have to do with jargon. Every doctor knows that in theory, you obviously should use layman's terms, that sort of thing. . . If you, as a medical professional, year*

*after year, connect separate thoughts or concepts in medical practice with each other [e.g., connecting symptoms with types of medication], or find certain things logical, that are, on a certain level, not that logical for patients, I think that we easily lose sight of that as doctors, and that is, obviously, extra difficult for patients with LHL. But I'll try to watch out for this. I sometimes notice this when supervising others, such as doctors' assistants, for example, who are at the beginning of their career. When they explain to the patient what their condition is and why we prescribe treatment, I sometimes notice they are telling something a patient really cannot understand. That's when you hear that it is really still too complicated.*

Some HCPs assess the appropriate moment to discuss palliative care by carefully introducing this topic to the patient, if the patient refuses to engage, HCPs state that it is important to underline the possibility of discussing this topic later on (Quote 13).

*Quote 13. Specialized nurse palliative care.*

*Respondent: If you already mention 'the end' a couple of times, later on, you can say, "I mentioned 'the end' earlier on, and by that, I mean 'death'". "Have you ever thought about that, about dying?" This is how I actually start the conversation. That's how you open up the conversation. Sometimes you'll notice that people, significant others, find it very difficult to talk about death, but it is oh so important, because if they don't do that, they are going to miss out on something. As soon as the conversation is opened-up, they are ready to take the next step. I think that this is actually one of the most important tasks I have during conversations with people who are moving towards the end of life. I can have them take the next step, so that the lump in their throats is gone, and they are ready to enter the last part of life with peace of mind, enabling them to find the things they think are important and pay attention to them, without being afraid of death.*

Furthermore, some HCPs *sensitize* their communication by empathizing with the patient. They ask patients about their feelings and emotions and actively describe the emotions patients express during consultations. According to these HCPs, by describing and acknowledging the emotions patients express, they make them part of the consultation and therefore more negotiable. Additionally, some HCPs create 'openness' by showing a genuine interest in the social lives of patients or ask questions related to their feelings. In addition, HCPs offer space for patients by allowing moments of silence during the consultation to stimulate patient participation and leave the initiative to talk with the patient. Some HCPs indicated that if rapport building with a patient is difficult, focusing on the experiences and feelings of significant others could indirectly create the desired level of understanding. Additionally, according to HCPs, being honest and straightforward is valued by patients. Regarding SDM, treatment options should be communicated within the context and preferences of the patient and significant others.

Additionally, HCPs recommend a *comprehensive and supportive interaction* with their patients. This means that HCPs recommend to openly discuss difficult subjects with patients and share the responsibilities of SDM when difficult choices are needed. They also state that refraining from treatment should be an option to discuss openly with their patients. HCPs should additionally support patients in considering this option, just as much as they support patients with other options—sharing responsibilities -, and that 'not treating' does not mean 'not doing anything' (Quote 14).

*Quote 14. Internist and consultant palliative care.*

*Respondent: . . . Your own opinion is in there, no matter how you look at it. You can try not to have an opinion, but you can't. Not everyone wants to have a choice. "Do whatever you like doctor". Sometimes I don't think it's fair to people to position them with an option. As an example, from my own personal life, my mother had to decide for her severely mentally and physically handicapped brother whether to continue his life with tube feeding, yes, or no. And she did call me. And I said, "stop [i.e., no tube feeding], because this doesn't contribute to the quality of his life". But my mother was the one who had to make that decision. She still feels that, well, not that she killed her brother. . . Of course, people have to make their own decisions. But at some point, they are not supposed to feel that they are solely responsible for making life or death decisions. So, yes, SDM, however, without withdrawing from your responsibilities as a doctor.*

Also, HCPs expressed the wish to discuss crisis situations and advanced care planning more comprehensively with patients. They believed that could avoid unnecessary hospital admissions. Furthermore, and in addition to empathizing with the patient, some HCPs would like to ask patients about their feelings and emotions. They suggested to discuss the lifeworld and goals of patients more empathically, and ask patients about their own future, for instance, "a year from now, how would you describe your own condition?". According to these HCPs, this facilitates 'real-talking' and effective communication, however, in most cases, takes up too much time in a consultation.

Lastly, HCPs tailor their communication by *repeating* messages and summarizing information regarding medication and treatment options during and at the end of the consultation.

**4. Characteristics of patients and significant others.**   According to the HCPs, barriers to effective communication can occur as a result of characteristics of patients and significant others. According to HCPs, all patients in general have a responsibility to improve communication; they suggest that patients have to take more initiative, prepare the consultation beforehand, think about the questions that the HCP is going to ask during the consultation and ask questions if they do not understand the information. However, despite the responsibilities patients should have, HCPs additionally indicate that some of their patients are not able or ready to talk about palliative care (Quote 15).

*Quote 15. Pulmonologist.*

*Respondent: With the vast majority of my patients, I cannot say "I don't think you will be here in a year", because then there will be no more conversation. [. . .]. The conversation blocks. So sometimes, after a consultation, a significant other returns to the room when grandpa has already left for the counter, saying, "what do you think, doc, how is he?" And if I, at that moment, mention "I think he's gone in a year's time", the significant other will respond "I know and I can see that, but how should we continue?" Well, that's nice, because now you have an opening. However, if grandpa remains in the room and prefers to be left alone and wants to avoid confrontation . . . A lot of people don't want to know that at all [whether they are still here in a year's time].*

More specifically, the *knowledge* of patients can be a barrier in communication. According to HCPs, many patients with LHL lack knowledge about anatomy or how the human body functions, are unable to understand treatment options, or do not recognize an emergency situation. In addition, according to some HCPs, patients in general in palliative care—but especially patients with LHL in palliative care—have a hard time of deciding when to stop undergoing treatment (Quote 16).

*Quote 16: Specialized nurse in oncology.*

*Respondent: Well, the thing I often notice, and I don't know if that's because of LHL, but I do think so, is that people in the palliative phase with LHL often want to continue treatment for a very long time. They only want to treat, treat, treat. And at some point, they are incapable of saying, well, that's enough. My chances are slim to none, so now we have to stop. And that is an important problem.*

*Interviewer: Okay. And why do you think that is? Why is that the case for people with LHL in particular?*

*Respondent: I think they are unable to understand those decisions. Look, it all comes down to probability calculations, right? If we give chemotherapy, there is a 2% chance that you'll be alive in five years, compared to not giving chemotherapy. And even for us [people not having LHL] this is complicated. If you don't even understand what a percent is and these probabilities are depicted on yourself, obviously, then it is impossible to form an opinion about that. And that creates a mode of survival for people, meaning treatment must go on. At all cost.*

Furthermore, HCPs indicated that the *attitude*, *mood or condition* of the patient with respect to palliative care can be a barrier. Some patients can be very closed, stiff, distant, or only use very short sentences. Or they are upset, emotional or angry. According to a few HCPs, if patients feel anxious about receiving treatment, they tend to lie about the amount of pain they feel or the actual condition they are in.

Lastly, a divergent *language and culture* is perceived as a barrier in communication and SDM. For example, HCPs indicated that they are not able to check what a significant other or interpreter is communicating to the patient, which probably further decreases mutual understanding.

**5. Content of medical information.**   The complex nature of decisions that patients have to make in palliative care, for instance, deciding on permitting Intensive Care (IC) admissions or 'do not resuscitate' statements, are, according to HCPs, barriers for SDM. HCPs indicate that this is not only caused by the substantive complexity of the information, for instance, dealing with 'chance of survival statistics' that are too complex for patients to understand, but that the information is too complex for patients to imagine—i.e., matching experiences with expectancies. According to HCPs, patients are not able to imagine the options they are bound to choose from. Furthermore, according to some HCPs, discussing topics related to palliative care could provide patients with a sense of hopelessness that could impede communication.

## Discussion

This study assessed strategies, barriers and suggestions of HCPs regarding communication and SDM with LHL patients in hospital-based palliative care. Representing a prominent theme, and according to HCPs, more time to communicate with their patients could resolve the most prominent barriers emerged from this study. Planning sufficient time for a consultation by HCPs is also found by Chou and colleagues as a strategy to support patients with LHL in palliative care [34]. Furthermore, in a recent knowledge synthesis exploring challenges of HCPs with LHL in curative care, not enough time is mentioned by HCPs as a reason for not considering LHL during consultations [29]. In another knowledge synthesis investigating experiences of patients, the patients considered 'enough time' as a precondition for effective SDM, and in their experiences mentioned a lack of time as a barrier [30]. Although more time could possibly resolve barriers, the results of this study showed that when HCPs are

planning more consultation-time, this generates tension between HCPs and hospital-management. Therefore, further research should investigate the organizational opportunities of creating more communication-time between patients and HCPs and in general, the possible influence of more time on effectiveness of communication and SDM between patients and HCPs.

Additionally the results of this study indicated that for most HCPs, LHL is a relatively new concept. As mentioned, a majority of the HCPs had never heard of limited *health* literacy prior to the interviews. Nevertheless, after defining LHL during the interviews, all participating HCPs did acknowledge the importance of paying attention to LHL in palliative care, and realized they see LHL patients on a daily basis. This finding is reflected in the knowledge syntheses investigating challenges of HCPs with LHL in curative care [29]. In addition, all HCPs provided helpful strategies and suggestions for effective communication and SDM with patients with LHL. Hence, it seems that HCPs, despite not knowing LHL as a concept, do adapt their communication to the charateristics of their patients having LHL to some degree. However, HCPs could benefit from knowing and recognizing LHL, and use strategies that are helpful to patients, increasing understanding, communication and SDM [28–30]. Perhaps, HCPs could find comfortable ways of eliciting conversations about the LHL-level of patients by asking them a screening question using the single item literacy screener (SILS) [35]. Additionally, in order to be helpful to patients with LHL, HCPs could refer patients to language training courses, or underline the positive and supportive role of family and social networks to seek, understand and use health information [36].

Furthermore, according to HCPs, effective communication and SDM is only achievable if there is a level of understanding or 'bond' with the patient and their significant others. However, despite being able to bond and discuss profound aspects of care, HCPs did observe a lack of skills by LHL-patients to understand complex medical information. This is also found in previous research [37], and insufficient understanding of the illness trajectory has previously been found to be a barrier to SDM in palliative care [30,38]. Furthermore, previous research has indicated that HCPs find it difficult to address advanced care planning with LHL-patients [39]. This could indicate that the lack of skills by patients and HCPs, and the complexity of medical information both determine effective communication and SDM, despite the relationship and the degree of bonding between HCP and patient.

The lack of supportive materials that help HCPs to explain and communicate medical information to patients is another barrier reported by HCPs, and is also found in earlier research [26]. Educational material could improve the communication and SDM with patients with LHL [40,41]. Currently, there are visual communication materials available online with understandable health information (e.g., [28–30,42]. However, HCPs are usually not aware of the existence of these materials. Furthermore, in the present study, HCPs mentioned communication training and receiving feedback on their own communication as ways to improve their skills [38].

Since HCPs in this study had never heard of limited *health* literacy prior to the interviews, increasing awareness is important. This is underlined by similar research in other settings [28]. Creating more awareness about LHL and making sure HCPs know what LHL is, could be an important precondition for recognizing LHL and improving communication and SDM with this group of patients. Also, future research should investigate the ways to effectively tailor communication to patients with LHL. In addition, results from this study point to an ineffective use of the teach-back method. Despite its proven effectiveness for checking the understanding of patients with LHL [28,29], this study underlines the need of training HCPs to rightly and effectively apply this method.

## Strengths and limitations

To the best of our knowledge, this qualitative study focusing on HCPs and patients with LHL in palliative care is the first of its kind. No comparable studies have been performed concerning communication challenges related to patients with LHL in hospital-based palliative care, investigating the perspectives and unique experiences of HCPs. There is research available about barriers regarding communication with patients with LHL, however, this research was performed outside the palliative care setting [29,43,44]. As an additional asset of our study, some of our findings go beyond communication with people with LHL, addressing relational and communication issues more general within palliative care (e.g., in our second theme; 'HCPs' communication skills'). In addition, some limitations are important to note. Interviews were transcribed verbatim, excluding verbal utterances and pauses. Cues on the way in which things were said and how the participants were feeling were difficult to extract from the transcripts. Furthermore, the age of HCPs and years of experience as HCP were not registered. In this study, only experiences and views of HCPs are included. In future research, the experiences and views of patients with LHL in palliative hospital-based care should be investigated as well, perhaps considering visually augmented interviewing techniques using photo elicitation or photovoice, in which visual imagery is used to evoke feelings and experiences, to further enrich study results [45,46].

## Conclusions

This study provides first insights into the experiences of HCPs, indicating directions for further research on communication, SDM and LHL in hospital-based palliative care. HCPs experienced several barriers in effective communication with patients with LHL. As the concept of limited *health* literacy was unknown to most of the interviewed HCPs, more knowledge and awareness of LHL should be created. Furthermore, HCPs should receive training to recognize LHL, to adjust their communication to LHL-patients, and to facilitate patients to engage in SDM. Hospitals should look into increasing the length of consultations with patients with LHL, and support the development and implementation of new forms of training and existing and new visual educational materials.

## Supporting information

**S1 Appendix. Visual overview of the project 'A Basic Understanding'.**
(DOCX)

**S2 Appendix. Interview topic list.**
(DOCX)

**S1 Text.**
(DOCX)

**S2 Text.**
(DOCX)

**S3 Text.**
(DOCX)

## Acknowledgments

The authors thank all participating healthcare providers for their valuable contributions to the study.

## Author Contributions

**Conceptualization:** Ruud Roodbeen, Astrid Vreke, Jany Rademakers, Maria van den Muijsenbergh, Janneke Noordman, Sandra van Dulmen.

**Data curation:** Ruud Roodbeen, Astrid Vreke.

**Formal analysis:** Ruud Roodbeen, Astrid Vreke, Sandra van Dulmen.

**Funding acquisition:** Gudule Boland, Jany Rademakers, Maria van den Muijsenbergh, Janneke Noordman, Sandra van Dulmen.

**Investigation:** Ruud Roodbeen.

**Methodology:** Ruud Roodbeen, Gudule Boland, Jany Rademakers, Maria van den Muijsenbergh, Janneke Noordman, Sandra van Dulmen.

**Project administration:** Ruud Roodbeen, Gudule Boland, Janneke Noordman.

**Resources:** Ruud Roodbeen, Janneke Noordman.

**Software:** Ruud Roodbeen, Astrid Vreke.

**Supervision:** Jany Rademakers, Maria van den Muijsenbergh, Janneke Noordman, Sandra van Dulmen.

**Validation:** Ruud Roodbeen.

**Writing – original draft:** Ruud Roodbeen, Astrid Vreke.

**Writing – review & editing:** Ruud Roodbeen, Astrid Vreke, Gudule Boland, Jany Rademakers, Maria van den Muijsenbergh, Janneke Noordman, Sandra van Dulmen.

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
