## [Decision Letter · Decision Letter 0]

29 Apr 2020

PONE-D-20-02928

Communication and shared decision-making with patients with limited health literacy; helpful strategies, barriers and suggestions for improvement reported by hospital-based palliative care providers

PLOS ONE

Dear Dr Roodbeen,

Thank you for submitting your manuscript to PLOS ONE. After careful consideration, we feel that it has merit but does not fully meet PLOS ONE’s publication criteria as it currently stands. Therefore, we invite you to submit a revised version of the manuscript that addresses the points raised during the review process.

Please see comments below. 

We would appreciate receiving your revised manuscript by 28 May 2020. To enhance the reproducibility of your results, we recommend that if applicable you deposit your laboratory protocols in protocols.io, where a protocol can be assigned its own identifier (DOI) such that it can be cited independently in the future. For instructions see: http://journals.plos.org/plosone/s/submission-guidelines#loc-laboratory-protocols

We look forward to receiving your revised manuscript.

Kind regards,

Andrew Soundy

Academic Editor

PLOS ONE

Journal Requirements:

3. Thank you for stating the following in the funding Section of your manuscript:

"This study was funded by a grant of the Netherlands Organisation for Health Research and Development, Palliative Care Program - ZonMw, Palliantie: 844001403."

"Yes.

The financial disclosure statement is attached as a PDF file, including

signatures/initials of authors who received funding, the name and grant number of the

funder and the role description of the funder."

Additional Editor Comments (if provided):

Please consider the comments from the reviewers.

Please consider the COREQ (Tong et al., 2007) OR SRQR (Obrien et al., 2014) for examples of how to set out the methods section. This could include the need for you paradigmatic position and make sure this links to you section on quality certain philosophical stances for instance may not consider member checking as needed. It could include items like sample size justification (I have no issue with the sample size - justification can be added for the reader). Can you include in a supplementary file your initial interview schedule and the final version following piloting. Can you also include an audit trail of the analysis steps but just include examples of what you have done so the reader can see how you have gone from text to your analysis final version.

Reviewers' comments:

Reviewer's Responses to Questions

**Comments to the Author**

1. Is the manuscript technically sound, and do the data support the conclusions?

Reviewer #1: Yes

Reviewer #2: Yes

2. Has the statistical analysis been performed appropriately and rigorously? 

Reviewer #1: Yes

Reviewer #2: N/A

3. Have the authors made all data underlying the findings in their manuscript fully available?

Reviewer #1: Yes

Reviewer #2: Yes

4. Is the manuscript presented in an intelligible fashion and written in standard English?

Reviewer #1: Yes

Reviewer #2: Yes

5. Review Comments to the Author

Reviewer #1: I have some comments to be considered for this paper.

line 161 the information on the exclusion of three interviews in the calculation was a bit confusing, I'm not sure what you meant by the duration was not recorded. I thought that the audio recording would give the length of time of each interview.

Line 165 - I did not fully understand the purpose of member checking the transcripts for accuracy. Were the audio-recordings transcribed verbatim? Line 164 says the the were transcribed verbally. If they were transcribed verbatim then I'm not sure I understand why they were checked for accuracy.

Line 181 - I felt the description of some and few maybe unnecessary.

Line 225 I think in your results section you should introduce your table of themes before you talk about each of them. I'm not sure if that is a formatting problem but your table needs to be inserted before 1. Time management

In general you could reflect in your discussion that maybe clinicians can find comfortable ways of eliciting conversations about patient's health literacy level by asking them a screening question. There is single item health literacy screening tool available (SILS) Morris et al 2006. It is possible that clinicians can have a role of helping patients develop health literacy through their information exchange in consultations or that health literacy can be 'distributed' in one's social network. Therefore, more health literate family members may be able to support patients with health literacy. Edwards et al 2015 -'Distributed health literacy': longitudinal qualitative analysis of the roles of health literacy mediators and social networks of people living with a long‐term health

Reviewer #2: Congratulations on submitting this strong paper. I think that it will make a good contribution to the filed of communication in palliative care decision making. I found the data to be intriguing and very compelling and in many ways going beyond the original scope of LHL, but to include communication and relational practices more widely. I have just a few minor commentss to improve the paper still further:

1) The aim of the study on page three could be clearer- please look again at this.

2) The sequencing and placement of Table 2 needs attention-place a bit earlier in the manuscript please.

3) It is a shame that you did not collect the years of experience of each of the participants because this would have been useful contextual information. Is there a way that this could be accessed and included in Table 1?

4) I think at times the paper goes beyond communication with people with LHL. Is it possible in the dicssuion to state this? Theme 2 in particular addresses relational and communication issues more generally within palliative care and I think you should say this.

5) In the strengths and limitations section you use the phrase 'disabling theoretical sampling' -I'm not what this means, but in any case you did not set out to undertake TS so you can remove this.

6. PLOS authors have the option to publish the peer review history of their article (what does this mean?). If published, this will include your full peer review and any attached files.

Reviewer #1: No

Reviewer #2: Yes: Tony Ryan

---

## [Author Response · Author response to Decision Letter 0]

26 May 2020

Response to (editor and) Reviewers

Thank you for stating the following in the funding Section of your manuscript:

"This study was funded by a grant of the Netherlands Organisation for Health Research and Development, Palliative Care Program - ZonMw, Palliantie: 844001403."

"Yes.

The financial disclosure statement is attached as a PDF file, including

signatures/initials of authors who received funding, the name and grant number of the

funder and the role description of the funder."

We have removed all funding-related texts from the manuscript, as requested. 

Is it perhaps possible to add the sentence below to the Funding Statement registration? 

"This study was funded by a grant of the Netherlands Organisation for Health Research and Development, Palliative Care Program - ZonMw, Palliantie: 844001403."

Additional Editor Comments

Please consider the comments from the reviewers.

Please consider the COREQ (Tong et al., 2007) OR SRQR (Obrien et al., 2014) for examples of how to set out the methods section. This could include the need for you paradigmatic position and make sure this links to you section on quality certain philosophical stances for instance may not consider member checking as needed. It could include items like sample size justification (I have no issue with the sample size - justification can be added for the reader). 

We have restructured the method section of our manuscript following the domains and points of attention described in the COREQ checklist (Tong et al., 2007), as requested by the editor. In addition to restructuring, we have added new information to the method section, in which we describe the research team and design of our study more profoundly. The restructured and improved method section is added to the revised manuscript. Hopefully, these improvements will resolve the concerns voiced by the editor. 

Can you include in a supplementary file your initial interview schedule and the final version following piloting? 

We have added the first conceptual interview schedule and the final version (after piloting) in a supplementary file (named ‘Supplementary file – interview schedules’). Unfortunately, these versions are in Dutch (as well as the transcripts used in this study that are provided to the journal and the additional trail of steps in analysis described later on, as requested by the editor). To accommodate all international readers of the journal, we have added translations to the questions prompted in the interview schedules. Also, we have added comments to address key differences between the initial and definitive versions of the interview schedules. In addition, we have added some contextual information to the interview schedule in the manuscript of our study. Hopefully, this will resolve this comment of the editor. 

Can you also include an audit trail of the analysis steps but just include examples of what you have done so the reader can see how you have gone from text to your analysis final version.

Following the steps described in our manuscript (in the revised analysis section of our ‘Methods and materials’ paragraph), we have added examples of what we have done, leading up to our final version presented in the results section of our manuscript. We have described this information in a supplementary file (named ‘Supplementary file – steps of analysis’). All screencaps or images used in describing the steps in analyses are in Dutch. To accommodate all international readers of the journal, we have added translations to the main categories when applicable. In addition, if readers of PLOS ONE have questions regarding our study, we are more than willing to communicate directly with them. Perhaps, we could share the final version of our MAXQDA analysis file and provide a walk-trough of our analysis with it. Contact details of the corresponding author are provided in the resubmitted manuscript. Hopefully, this will resolve this remark of the editor. 

Regarding all remarks by the editor, we strongly believe that, by providing all transcripts, topic scheme versions and elaborate steps of analyses, replication of our study is possible. 

Comments to the Author

1. Is the manuscript technically sound, and do the data support the conclusions?

Reviewer #1: Yes

Reviewer #2: Yes

2. Has the statistical analysis been performed appropriately and rigorously? 

Reviewer #1: Yes

Reviewer #2: N/A

3. Have the authors made all data underlying the findings in their manuscript fully available?

Reviewer #1: Yes

Reviewer #2: Yes

4. Is the manuscript presented in an intelligible fashion and written in standard English?

Reviewer #1: Yes

Reviewer #2: Yes

5. Review Comments to the Author

Reviewer #1: I have some comments to be considered for this paper.

line 161 the information on the exclusion of three interviews in the calculation was a bit confusing, I'm not sure what you meant by the duration was not recorded. I thought that the audio recording would give the length of time of each interview.

The main author of this study removed the audio-files of the interviews and neglected to doublecheck for duration of the interviews prior to removing. After removing the audio-files, we found out that the transcription service did not register all the durations of the interviews on the manuscripts, they missed three of them. Therefore, three were excluded in the calculation. We have adjusted the sentence mentioned by the reviewer in the manuscript, hopefully clarifying this: 

The interviews took on average 46 minutes, ranging from 33 to 70 minutes. Three were excluded in this calculation, since the duration was accidentally not recorded by the researcher (RR).

Line 165 - I did not fully understand the purpose of member checking the transcripts for accuracy. Were the audio-recordings transcribed verbatim? Line 164 says the the were transcribed verbally. If they were transcribed verbatim then I'm not sure I understand why they were checked for accuracy.

The interviews were transcribed verbatim, we have changed the word ‘verbally’ with ‘verbatim’ in the manuscript, as suggested by the reviewer. Furthermore, we mention in the manuscript that this member check increases the credibility, accuracy and authenticity of the transcripts. The mentioning of ‘accuracy’ (as stated by the reviewer) is indeed unnecessary and irrelevant, since interviews are transcribed verbatim. We have removed the word ‘accuracy’ from the sentence in the manuscript, as suggested by the reviewer. 

Line 181 - I felt the description of some and few maybe unnecessary.

We agree with the reviewer and have removed this description from the manuscript. 

Line 225 I think in your results section you should introduce your table of themes before you talk about each of them. I'm not sure if that is a formatting problem but your table needs to be inserted before 1. Time management

We agree with the reviewer, and have inserted the table before the first theme (1. Time management). 

In general you could reflect in your discussion that maybe clinicians can find comfortable ways of eliciting conversations about patient's health literacy level by asking them a screening question. There is single item health literacy screening tool available (SILS) Morris et al 2006. It is possible that clinicians can have a role of helping patients develop health literacy through their information exchange in consultations or that health literacy can be 'distributed' in one's social network. Therefore, more health literate family members may be able to support patients with health literacy. Edwards et al 2015 -'Distributed health literacy': longitudinal qualitative analysis of the roles of health literacy mediators and social networks of people living with a long‐term health

We thank the reviewer for this interesting and relevant suggestion. In our research we have indeed used such a single item HL screening tool: ‘We know many people find it difficult to fill in a form. How about you?’ We have added the SILS to our discussion paragraph as an option for HCPs to elicit LHL in the sentence described below: 

Perhaps, HCPs could find comfortable ways of eliciting conversations about the LHL-level of patients by asking them a screening question using the single item literacy screener (SILS) (35). 

Furthermore, we have added the importance of HCPs distributing information beyond the consultation in the hospital and into the social networks of patients and significant others as well, including the referral of a patient to a language training course. We have added the sentence below to the manuscript: 

Additionally, in order to be helpful to patients with LHL, HCPs could refer patients to language training courses, or underline the positive and supportive role of family and social networks to seek, understand and use health information (36).

Reviewer #2: Congratulations on submitting this strong paper. I think that it will make a good contribution to the field of communication in palliative care decision making. I found the data to be intriguing and very compelling and in many ways going beyond the original scope of LHL, but to include communication and relational practices more widely. 

We thank the reviewer for this compliment. 

I have just a few minor comments to improve the paper still further:

1) The aim of the study on page three could be clearer- please look again at this.

We agree with the reviewer that aim of the study could be described more clearly. We have adjusted the section mentioned by the reviewer in the manuscript, hopefully clarifying this: 

Little is known about strategies, barriers or suggestions HCPs have with regard to communication and SDM in hospital-based palliative care, and especially so with patients with LHL. For this reason, the aim of this study is to investigate exactly that. Insights could offer HCPs recommendations to adjust their communication to the needs and preferences of patients, checking understanding and enhancing recall of medical information by patients. 

2) The sequencing and placement of Table 2 needs attention-place a bit earlier in the manuscript please.

We agree with the reviewer, and have inserted the table before the first theme (1. Time management).

3) It is a shame that you did not collect the years of experience of each of the participants because this would have been useful contextual information. Is there a way that this could be accessed and included in Table 1?

We agree with the reviewer that not collecting the years of experience of the participants is a shortcoming of our study. Unfortunately we did not collect and include this contextual information to the manuscript. However, we do mention this important shortcoming in the discussion paragraph of the manuscript, as described below: 

Furthermore, the age and years of experience as HCP were not registered. 

4) I think at times the paper goes beyond communication with people with LHL. Is it possible in the discussion to state this? Theme 2 in particular addresses relational and communication issues more generally within palliative care and I think you should say this.

We thank the reviewer for this interesting and relevant suggestion. It is true that our study goes beyond communicating with people with LHL (especially in theme 2), and this should be mentioned in the discussion paragraph, as suggested by the reviewer. We have added the sentence below to the discussion paragraph of our manuscript:

As an additional asset of our study, some of our findings go beyond communication with people with LHL, addressing relational and communication issues more general within palliative care (e.g., in our second theme; ‘HCPs’ communication skills’).

5) In the strengths and limitations section you use the phrase 'disabling theoretical sampling' -I'm not what this means, but in any case you did not set out to undertake TS so you can remove this.

We agree with the reviewer, and have removed the sentence from the manuscript. 

6. PLOS authors have the option to publish the peer review history of their article (what does this mean?). If published, this will include your full peer review and any attached files.

Do you want your identity to be public for this peer review? For information about this choice, including consent withdrawal, please see our Privacy Policy.

Reviewer #1: No

Reviewer #2: Yes: Tony Ryan

---

## [Editor Report · Decision Letter 1]

5 Jun 2020

Communication and shared decision-making with patients with limited health literacy; helpful strategies, barriers and suggestions for improvement reported by hospital-based palliative care providers

PONE-D-20-02928R1

Dear Dr. Roodbeen,

We’re pleased to inform you that your manuscript has been judged scientifically suitable for publication and will be formally accepted for publication once it meets all outstanding technical requirements.

Kind regards,

Katie MacLure, PhD, MSc (dist)., BSc (1st)

Academic Editor

PLOS ONE
---

## [Editor Report · Acceptance letter]

11 Jun 2020

PONE-D-20-02928R1 

Communication and shared decision-making with patients with limited health literacy; helpful strategies, barriers and suggestions for improvement reported by hospital-based palliative care providers 

Dear Dr. Roodbeen:

I'm pleased to inform you that your manuscript has been deemed suitable for publication in PLOS ONE. Congratulations! Your manuscript is now with our production department. 

Kind regards, 

on behalf of

Dr. Katie MacLure 

Academic Editor

PLOS ONE